# First Latin American Case of MLASA2 Caused by a Pathogenic Variant in the Anticodon-Binding Domain of *YARS2*

**DOI:** 10.3390/ijms262412039

**Published:** 2025-12-14

**Authors:** José Rafael Villafán-Bernal, Jhonatan Rosas-Hernández, Humberto García-Ortiz, Angélica Martínez-Hernández, Cecilia Contreras-Cubas, Israel Guerrero-Contreras, Hane Lee, Go Hun Seo, Alessandra Carnevale, Francisco Barajas-Olmos, Lorena Orozco

**Affiliations:** 1Immunogenomics and Metabolic Diseases Laboratory, Instituto Nacional de Medicina Genómica, Secretaría de Salud, Mexico City 14610, Mexicohgarcia@inmegen.gob.mx (H.G.-O.); amartinez@inmegen.gob.mx (A.M.-H.); ccontreras@inmegen.gob.mx (C.C.-C.); israel.guerrerocon@gmail.com (I.G.-C.); 2Programa de Investigadoras e Investigadores por México, Secretaría de Ciencia, Humanidades, Tecnología e Innovación, Mexico City 03940, Mexico; 3Centro de Rehabilitación Infantil Teletón, Altamira 89607, Tamaulipas, Mexico; dr.rosas.genetica@outlook.com; 43billion, Inc., Seoul 03161, Republic of Korea; hlee@3billion.io (H.L.); ghseo@3billion.io (G.H.S.); 5Mendelian Diseases Laboratory, Instituto Nacional de Medicina Genómica, Secretaría de Salud, Mexico City 14610, Mexico; acarnevale@inmegen.gob.mx

**Keywords:** MLASA2, *YARS2*, meta-analysis, sideroblastic anemia, lactic acidosis

## Abstract

MLASA2 is a rare mitochondrial disorder with limited geographic representation in published medical literature. Here, we report the first confirmed case of MLASA2 in a Latin American 16-year-old male harboring a homozygous pathogenic variant p.(Asp311Glu) in the *YARS2* gene. The patient presented with sideroblastic anemia and short stature, accompanied by other skeletal dysplasia features not previously associated with MLASA2, including epiphyseal dysplasia, rib edge widening, and poorly defined vertebral structures, but without lactic acidosis. Notably, the patient did not present exercise intolerance but recently exhibited reduced muscle strength. The p.(Asp311Glu) variant, located in the anticodon-binding domain of the mitochondrial tyrosyl-tRNA synthetase (Mt-TyrRS), was consistently predicted to be pathogenic by multiple in silico tools. Molecular modeling revealed that this variant destabilizes the ‘KMSKS’ motif, potentially compromising tRNA recognition fidelity and aminoacylation efficiency. Analysis of runs of homozygosity (ROH) revealed significantly elevated consanguinity (ROH: 31.93%), consistent with a consanguineous mating between biological parents. This case expands the geographic distribution of MLASA2, documents previously unreported phenotypes, suggests a novel pathogenic mechanism, and demonstrates the utility of genomic approaches for diagnosing rare mitochondrial disorders in the absence of complete clinical information and family history.

## 1. Introduction

MLASA syndrome type 2 (MLASA2; OMIM # 613561) is a rare autosomal recessive mitochondrial disorder caused by pathogenic variants (PV) in the *YARS2* gene, which encodes the mitochondrial tyrosyl-tRNA synthetase (mtTyrRS) [1]. Since the initial description of the disease by Riley (2010) [2], less than 40 cases have been reported worldwide, with predominant representation from Western Europe, North America, and Australia, and many of them of Lebanese descent [3,4]. To our knowledge, no confirmed cases of MLASA2 have been previously reported in Mexico or Latin America.

MLASA2 manifests with marked phenotypic heterogeneity, and diagnostic challenges frequently result in diagnostic delay. Classic clinical features include myopathy, lactic acidosis, sideroblastic anemia, and exercise intolerance [5]. However, heterogeneous presentations have been documented. Some cases present with isolated anemia and no other evident clinical manifestations and prolonged survival [4], whereas others manifest severe decompensation of cardiorespiratory function and short survival [6].

Here, we report the first MLASA2 case in the Mexican population carrying a homozygous PV in *YARS2* gene. We also modeled the structural protein dysfunction caused by the pathogenic variant and analyzed runs of homozygosity (ROH) to assess for consanguinity due to the absence of biological parents.

## 2. Results

### 2.1. Clinical Case Presentation

The male case reported here was born at 35 weeks of gestation with an Apgar score of three at 1 min, five at 5 min, and eight after 10 min. He was adopted at 14 days of age; hence, his familial history was unavailable. Gross motor development included neck control at 3 months, trunk control at 6 months, crawling at 8 months, and walking at 18 months. Meanwhile, fine motor development began at 4 months and speech milestones at 2 years.

At 11 years old, the patient was referred to a Genetic Clinic presenting with epiphyseal dysplasia, metaphyseal widening and irregularity, widening of rib edges, poorly defined vertebral structures, scoliosis, and *genu valgum*. At the same age, laboratory tests revealed low alkaline phosphatase serum levels (119 U/L), lactate dehydrogenase [LDH] 140 U/L (at lower normal end; reference values 235–470), and slightly increased levels of serum phosphorus (5.9 mg/dL; reference < 4.5 mg/dL) with normal calcium and parathormone levels. The patient presented moderate-severe normocytic–normochromic anemia (Hb 7 g/dL), two bone marrow aspirates confirmed sideroblastic rings without evidence of myeloproliferative cause, and thrombocytopenia was detected (<100,000/μL).

At 13 years old, the patient exhibited short stature (percentile < 3), low hairline implantation, horizontal palpebral fissures, high nasal root, dorsal nasal hump with depressed tip, dental malocclusion and crowding, posteriorly rotated ears with hypoplastic antihelix and deep concha, limited elbow extension and pronation–supination, and left-predominant *genu valgum*. No ocular, neurological, cardiovascular, or respiratory signs and symptoms were identified during the physical examination. However, the patient presented with their most severe anemia episode (Hb 4 g/dL) requiring two transfusions. A clinical exome panel, including variants in 109 genes causing skeletal dysplasia, was performed, and the results were negative.

At 15 years old, the patient was referred to our institution, and Whole Exome Sequencing (WES) revealed a homozygous PV located in the *YARS2* gene [chr12:32906866 G>C (GRCh37); c.933C>A; p.(Asp311Glu)], found in the anticodon-binding domain of mtTyrRS.

At present, at the age of 16 years, the patient no longer presents with anemia (Hb = 12.5 g/dL) and is currently managed only with folic acid supplementation, demonstrating non-transfusion dependence; however, he continues to have mild thrombocytopenia (platelet count of 89,000/μL). Cardiological evaluation revealed no cardiomyopathy and serum lactate levels were 1.4 mmol/L (normal range 0.7–2.1). In the last evaluation (April 2025), the patient exhibited decreased muscle strength in the upper extremities (4/5 bilaterally) and lower extremities (right 4/5, left 3/5). However, neither intolerance to exercise, fatigue, nor tiredness were referred to during the interview. No muscle biopsy nor additional enzymatic study was accepted by the patient’s legal guardians. Imaging studies demonstrated left-sided acetabular–femoral dysplasia with neoacetabulum formation and the genu *varum* predominantly affecting the left knee (15 degrees versus 10 degrees on the right). The patient is currently scheduled to undergo osteotomy to correct the acetabulum–femoral deformity. No additional information regarding the 24 h food recall and the food frequency questionnaire could be obtained from the legal guardians. Since Ardissone et al. [7] described two Italian siblings harboring the same PV identified in our patient, and Riley reported another case [4], we present a comparison of their clinical manifestations in Table 1.

### 2.2. In Silico Prediction of Pathogenicity

In silico prediction tools classified the p.(Asp311Glu) variant as damaging (SIFT), probably damaging (PolyPhen-2), disease-causing (MutationTaster), deleterious (FATHMM), possibly pathogenic (M-CAP score), and pathogenic (MutPred2) [Table A1]. To deepen the impact of PV p.(Asp311Glu) on protein structure and stability, we evaluated the changes in interactions between residue 311 and near amino acids using DynaMut2 and PyMOL3.1. In wild-type *YARS2*, the D311 amino acid interacts with H280, R287, and Q290, contributing to the stability of the ‘KMSKS’ motif. This region is crucial for maintaining the fidelity of tRNA recognition and for the efficient catalysis of the aminoacylation reaction, helping to coordinate the transfer of the amino acid to its cognate tRNA [8]. In the mutated *YARS2* [311E], these interactions are lost due to the disruption of ionic bonds and van der Waals forces. It results in a loss of stability in the ‘KMSKS’ motif demonstrated with the parameters of the stability and flexibility of the protein [ΔΔG and ΔΔSVib, respectively] (Figure 1).

### 2.3. p.(Asp311Glu) PV Interpretation According to ACMG/AMP

According to the criteria of the American College of Medical Genetics and Genomics (ACMG) and the Association for Molecular Pathology (AMP), the p.(Asp311Glu) PV fulfills two strong criteria (PS1 and PS3), one moderate (PM2), and two supporting [PP3 and PS3_supporting], classifying it as pathogenic (Table 2). Functional evidence (PS3) comes from Ardissone’s study, which showed that carriers of the pathogenic variant p.(Asp311Glu) had reduced activity of complexes I, III, and IV of oxidative phosphorylation in muscle tissue [7].

### 2.4. Estimation of Runs of Homozygosity (ROH)

Since the patient’s biological parents were unavailable for genealogical evaluation, we estimated the ROH as a proxy for consanguinity, considering that an offspring of a first-degree parental relationship typically leads to ROH values that exceed 25%. The ROH value for the index case was 31.93%, significantly higher than that exhibited in the reference cohort of 2217 Mexicans (16.05 ± 2.59%) [Figure 2].

## 3. Discussion

In rare genetic disorders, the convergence of multiple clinical manifestations often presents a complex puzzle that challenges medical practitioners to uncover the specific diagnosis and the underlying pathophysiological mechanisms [9], especially in clinical scenarios where all clinical features are not explained by one syndrome or disease.

Although the patient analyzed here presented transfusion-dependent sideroblastic anemia and lactic acidosis, two of the main characteristics of MLASA2 [10], he did not present exercise intolerance, nor was suspected of having myopathy. Instead, the patient displayed skeletal abnormalities not previously reported in MLASA2, including epiphyseal dysplasia, widening of the metaphysis, irregularity and widening of rib cages, poorly defined vertebral structures, scoliosis, genu *valgum*, and elevated serum phosphorus. These clinical features initially suggested bone dysplasia, delaying the accurate diagnosis. After WES, the homozygous PV p.(Asp311Glu) in *YARS2* was identified, reaching the precise diagnosis, and suggesting a phenotypic expansion of MLASA2 toward some skeletal abnormalities, such as epiphyseal dysplasia, widening of the metaphysis, irregularity and widening of rib cages, and poorly defined vertebral structures, because scoliosis was previously reported by Riley in a patient harboring another PV [p.(Phe52Leu)]. An explanation for the skeletal phenotypes of the index case might be found in a study by Holzer et al., who demonstrated in mice that postnatal inactivation of the respiratory chain in chondrocytes disorganizes the growth plate, precipitating premature closure, metaphyseal/epiphyseal defects, and short stature [11]. However, we cannot exclude other genetic and environmental factors contributing to the patient’s skeletal phenotypes, as no whole-genome sequencing was performed, and we cannot rule out the exposure to other environmental factors that can disrupt growth plate function during the antenatal or early postnatal period, because of the limited information associated with the adoption of the patient.

The hemoglobin level in our patient [7 g/dL] was similar to that reported by Riley [4] in a Dutch male patient [6.6 g/dL] and by Ardissone et al. [7] in an Italian male [5.2 g/dL], but lower than that of the latter’s sister [10.5 g/dL], suggesting the influence of additional factors on anemia severity. Furthermore, although our patient presented with anemia during adolescence, as in the case reported by Riley at 13 years old, the two cases reported by Ardissone et al. [7] presented it during the first two months of life. A similar pattern was observed for lactic acidosis, which is absent in our index case, whereas in the Italian cases it occurred during infancy and in the Dutch patient at 13 years of age (Table 1). Other PVs in YARS2 exhibited the same extreme phenotypic differences in the timing of anemia and lactic acidosis onset. For example, some carriers of p.(Phe52Leu) present anemia during the first months of life [2], whereas others develop anemia at 14 years of age [4]. Also, although lactic acidosis is a universal feature of p.(Phe52Leu) carriers [2,4,12], it is present in only 50% of those with p.(Leu392Ser), suggesting that variable expressivity is not exclusive of the p.(Asp311Glu), and that it does not depend on the domain where the PV is located, since patients carrying PV in the anticodon-binding or the catalytic domain do no present lactic acidosis [3,4,5].

In our patient, hemoglobin levels normalized after transfusions and subsequent folic acid supplementation alone, suggesting that adequate folate availability may enhance compensatory erythropoiesis despite the underlying mitochondrial defect [13]. While this observation does not imply disease modification, it raises the possibility that folic acid could serve as a useful supportive cofactor in selected individuals with YARS2-related MLASA2. Folic acid is a cofactor required for the synthesis of purines and thymidylate, and thus for proper DNA and RNA production during rapid proliferation of erythroid precursors [14]. Even in sideroblastic anemias where mitochondrial dysfunction, and not folate deficiency, is the primary pathology, increased erythropoietic turnover can raise folate requirements, and supplementation may support more efficient maturation of red-cell precursors by optimizing nuclear division and reducing ineffective erythropoiesis. However, Riley reported two cases of patients with MLASA2 who did not respond to folic acid therapy [2,4], and current evidence on the efficacy of folic acid for congenital sideroblastic anemia is unclear. For example, whereas LeBlanc et al. reported amelioration of congenital sideroblastic anemia with the administration of glycine plus folic acid in animal models, the transfusion requirement was not resolved in three patients with PV in the *SLC25A38* gene [15]. Cox and colleagues supplemented pyridoxin plus folic acid in three subjects with congenital X-linked sideroblastic anemia, achieving hemoglobin control that relapsed after discontinuation of supplementation and remitted after re-administration of pyridoxin [16]. Thus, folate supplementation might be a co-adjuvant therapy in sideroblastic anemia to meet the high folate demand associated with increased erythropoiesis. In any case, future studies should identify the optimal therapy and cofactors for achieving an optimal response in congenital sideroblastic anemia caused by YARS2 mutations and determine whether the anemia is self-limited.

Even if at present our patient has no clinical evidence of lactic acidosis, fatigue, or low physical performance, we cannot discard the future presentation of clinically recognizable myopathy and lactic acidosis during resting or exercise as presented by the male Dutch patient carrying p.(Asp311Glu) PV and by other patients harboring p.(Leu61Val) and [p.Gly191Val];[p.(Ile454Serfs*10] PV whose carriers developed lactic acidosis only during exercise [4]. For this reason, a timely follow-up will be mandatory, and the prescription of riboflavin and coenzyme Q10 will be suggested to the patient’s primary care physician, as it could enhance muscle function by improving mitochondrial mass and electron flow in the mitochondrial respiratory chain [17,18].

Although we cannot demonstrate the mechanisms underlying heterogeneity in the clinical presentation, some factors may be involved, including modifier genes, differences in mitochondrial haplogroups, tissue-specific mitochondrial thresholds, and nutritional and environmental factors. Modifier genes are mechanisms proposed in other recessive diseases, such as cystic fibrosis, for explaining variable expressivity [19]. Mexicans, when compared to Italians and Dutch, have different mitochondrial haplogroups (mH) [A2, B2, C1, D1 in Mexicans and H, U, J, T, K in Caucasians] that might influence the baseline respiratory efficiency, ROS management, and compensatory capacity; however, differences in oxidative phosphorylation, ATP, and ROS generation have, up until now, only been proved in cell models of some mH (H, J, T) [20] and for some mitochondrial diseases such as Leber’s optic neuropathy whose risk is modified by mH [21]. Tissue-specific biochemical–metabolical mitochondrial thresholds are a well-established concept whereby different organs manifest dysfunction only when oxidative phosphorylation drops below a critical level [22]; hence, it is possible that during developmental transitions such as pubertal growth, the increased metabolic demands may unmask latent mitochondrial insufficiency, contributing to disease presentation in patients with mitochondrial diseases, as in our index case. On the other hand, nutritional factors might mask disease presentation, as the administration of riboflavin and coenzyme Q10 appears to contribute to a better clinical evolution in a boy harboring p.(Asp311Glu) PV [7].

Although it was impossible to construct the patient’s family tree, the exceptional high ROH (31.93%) suggests inbreeding, since the ROH value was higher than the threshold defined for first-degree relatives (25%) [23], exceeding even those reported in two Mexican indigenous groups where endogamy and consanguinity are frequent because of the small population size and geographic and cultural isolation [24].

The in silico structural analysis revealed previously unexplored potential mechanisms of pathogenicity associated with p.(Asp311Glu). Our prediction indicated that this PV decreased the stability of the TyrRS ‘KMSKS’, a crucial motif for maintaining the fidelity of tRNA recognition [8] and for the efficient catalysis of the aminoacylation reaction [25].

The p.(Asp311Glu) PV fulfills criteria for being classified as pathogenic (two strong criteria, one moderate, and two supporting criteria) according to the ACMG/AMP [26]. PS3 (strong) is fulfilled because a previously reported patient carrying the same YARS2 p.(Asp311Glu) variant underwent well-established functional studies demonstrating a clear deleterious effect on mitochondrial oxidative phosphorylation. Specifically, a markedly reduced activity of respiratory chain complexes I, III, and IV (13%, 18%, and 15% of mean control values, respectively), and Seahorse-based oxygraphy in fibroblasts revealed a profound reduction in maximal respiratory capacity (28% of controls) [7]. These findings provide direct experimental evidence that the p.(Asp311Glu) substitution disrupts mitochondrial bioenergetics, fully satisfying the ACMG/AMP definition of PS3. PS1 (strong) is fulfilled because the variant identified in our patient results in the same amino acid change as previously established pathogenic variants reported in individuals with MLASA2, indicating that this substitution has been unequivocally associated with disease in independent cases. Such facts, in combination with the robust in silico analysis supporting pathogenicity (PP3) and the absence in population databases, including gnomAD (PP3), suggest that p.(Asp311Glu) is pathogenic.

We acknowledge the limitations of our study, including the lack of a muscle biopsy to evaluate respiratory chain activity, as subclinical mitochondrial myopathy may be present despite clinical silence. Furthermore, the failure to study the patient’s biological family limits our ability to determine variant segregation and potential carrier status among relatives. Furthermore, the absence of detailed metabolic profiling (plasma organic acids, carnitine profile) limits the complete characterization of mitochondrial dysfunction.

## 4. Materials and Methods

This study was performed in accordance with the ethical standards of the Declaration of Helsinki (1975, revised 2024). Written informed consent was obtained from the patient’s legal guardians for participation in the study, for genetic testing, and for publication of anonymized clinical and genetic data. All identifying information has been removed to preserve patient confidentiality. 

A complete clinical evaluation of the index case was performed, focusing on clinical phenotypes and relevant symptoms. The genetic tree was not constructed because the biological parents of the affected individual were unavailable. The patient’s legal guardians did not consent to additional examinations, such as biopsy and enzymatic activity measurements, or to measurements of biomarkers of mitochondrial dysfunction, such as acylcarnitines, plasma amino acids, or CoQ10 levels.

WES was performed as previously described [27]. Briefly, genomic DNA was extracted from EDTA blood specimens using a standard protocol. Exome capture was performed using the xGen Exome Research Panel v2 (Integrated DNA Technologies, Coralville, IA, USA), and sequencing was performed using NovaSeq 6000 (Illumina, San Diego, CA, USA). Sequencing reads were uniquely aligned to the Genome Reference Consortium Human Build 37 (GRCh37) and Revised Cambridge Reference Sequence (rCRS) of the mitochondrial genome, generating 198.84 mean depth-of-coverage within the captured region. Approximately 99.10% of the targeted bases were covered to a depth of ≥20x. Variant interpretation was performed using EVIDENCE v.1 (https://3billion.io/variant-interpretation, 1 December 2022) [28] and GEBRA v.1 (https://3billion.io/gebra/, 1 December 2022), proprietary software to prioritize and interpret variants based on the guideline recommended by the American College of Medical Genetics and Genomics (ACMG) and the Association for Molecular Pathology (AMP) [29] in the context of the patient’s phenotype, relevant family history, and previous test results.

To assess the pathogenicity of variants we used several in silico tools, including SIFT v4.0.4 (https://sift.bii.a-star.edu.sg/, 1 August 2025), PolyPhen v2.0 (http://genetics.bwh.harvard.edu/pph2/, 1 August 2025), MutationTaster 2025 (https://www.mutationtaster.org/, 1 August 2025), Functional Analysis Through Hidden Markov Models [FATHMM] v2.3 (http://fathmm.biocompute.org.uk/, 1 August 2025), Combined Annotation Dependent Depletion [CADD] v1.7 (https://cadd.bihealth.org/, 1 August 2025), Mendelian Clinically Applicable Pathogenicity [M-CAP] score (http://bejerano.stanford.edu/mcap/, 1 August 2025), MutPred2 (http://mutpred.mutdb.org/, 1 August 2025), and Local Indicator of Statistical Dependence using Spatial Smoothing [LIST-S2] v.1 (https://list-s2.msl.ubc.ca, 1 August 2025).

Structural impact of the PV on the protein stability was evaluated selecting the best PDB available [2PID; obtained by X-ray crystallography] and was used to estimate the ΔΔG, ΔΔG ENCoM, and the ΔΔSVib in DynaMut2 (https://biosig.unimelb.edu.au/dynamut/, 1 August 2025). The model generated was visualized and customized in PyMOL3.1 (https://www.pymol.org/, 1 August 2025). DynaMut2 was run with default parameters, including force-field energy minimization for 5 ns at 300K.

Because of the unavailability of the patient’s biological parents, runs of homozygosity (ROH) were estimated using PLINK as previously described [24]. ROH was calculated in the patient, employing as a reference the Mexican DMS2 cohort of 2217 individuals [30]. The ROH cut-off values to estimate consanguineous mating were 25%, 12.5%, 6.25%, and 1.56%, from first to fourth degrees, respectively [23].

## 5. Conclusions

The present case highlights the usefulness of genomic tools, such as next-generation sequencing, for enabling precision diagnosis and personalized medicine, as they facilitate the detection of disease-causing mutations, especially when clinical manifestations do not fit into an evident disorder. This study expands the knowledge of MLASA2 syndrome by describing the first Latin American individual affected and potentially new skeletal dysplasia features that have not been previously reported. Additionally, we provide new mechanistic insights into pathogenicity and employ ROH-based estimates to evaluate consanguinity in the absence of biological parents.

## Figures and Tables

**Figure 1 ijms-26-12039-f001:**
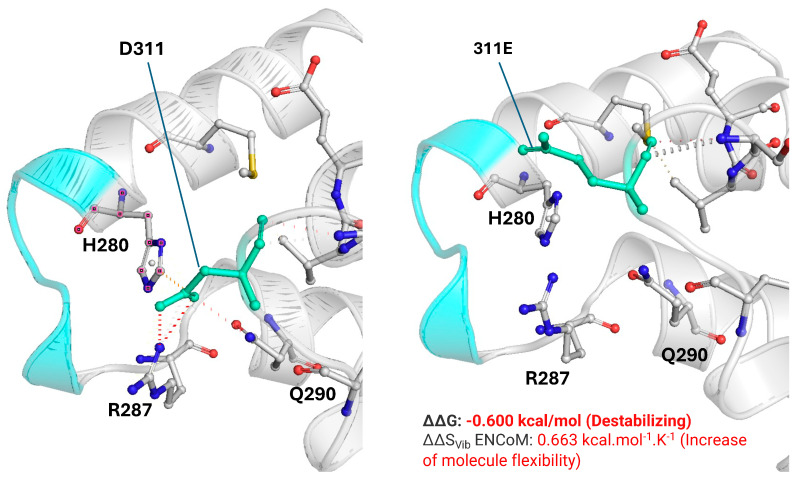
Structural comparison of the wild-type Asp311 and mutant Glu311 residues and their impact on the KMSKS motif of YARS2. Left: wild-type Asp311 (D311, green) within the anticodon-binding domain, showing its spatial relationship with the KMSKS loop (cyan) and neighboring residues His280, Arg287, and Gln290. Right: mutant Glu311 (311E, green) modeled at the same coordinates, displaying changes in the side chain that alter local packing and the interaction network around the KMSKS motif. DynaMut2 predicts a destabilizing effect for the p.(Asp311Glu) substitution with ΔΔG = −0.600 kcal/mol and an increase in molecular flexibility with ΔΔSVib ENCoM = +0.663 kcal·mol^−1^·K^−1^. The ΔΔG for other pathogenic variants such as p.(Gly46Asp) [catalytic domain], p.(Cys369Tyr) [anticodon-binding domain], and Gly244Ala [catalytic domain] are −0.84 kcal/mol, −0.82 kcal/mol, and −0.52kcal/mol, respectively.

**Figure 2 ijms-26-12039-f002:**
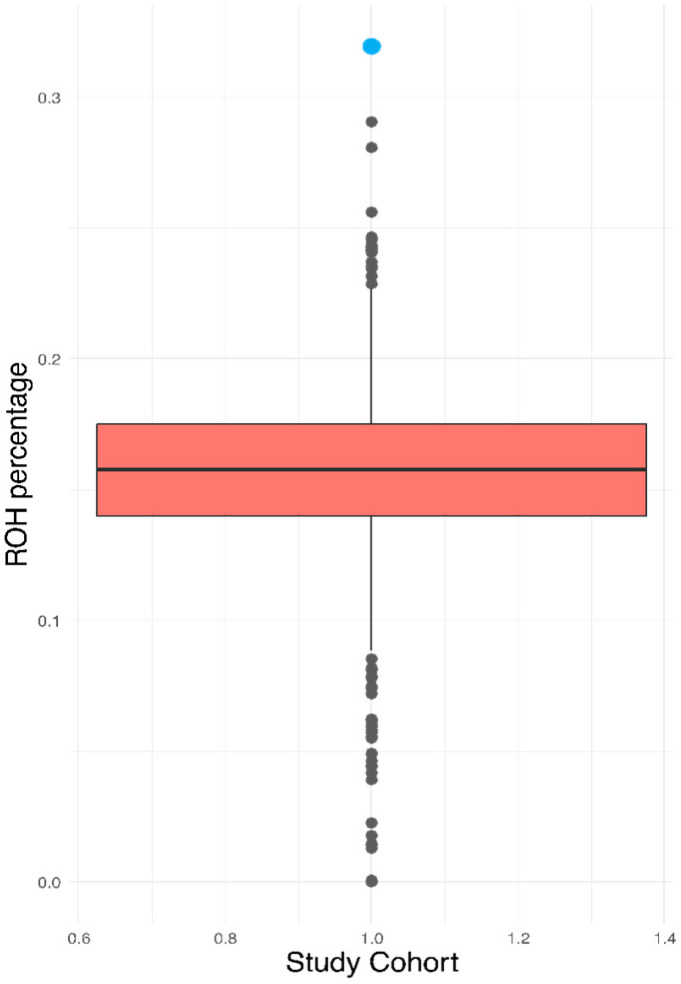
Graphical representation of the ROH values of the case [blue] and the total population.

**Table 1 ijms-26-12039-t001:** Comparative clinical features of MLASA2 cases (YARS2 p.(Asp311Glu)).

Domain	Feature	Mexican Index Case (Male)	Italian Sibling 1 (Male) [7]	Italian Sibling 2 (Female) [7]	Dutch (Male) [4]
Birth and Perinatal	Consanguinity; gestational age (GA); perinatal course.	No; 35 weeks; Apgar 3/5/8; adopted at 14 days.	No; GA at birth: NR; uncomplicated.	No; GA at birth: NR; uncomplicated.	No; GA at birth: NR; uncomplicated.
Development	Motor milestones	Neck 3 mo [N] Walk 18 mo [N]	Neck [N] Walk 18 mo [N]	Neck [N] Walk [N]	Neck [NR] Walk [NR]
	Fine motor/speech	[N]	NR	NR	NR
Growth	Stature	Short stature (<P3)	P50	NR	[N]
Hematological	Age at anemia onset [Hb]	11 y [7 g/dL] The lower 4 g/dL	2 mo [5.2 g/dL]	1 mo [10.2 g/dL]	13 y [6.6 g/dL]
	Transfusion dependence	No	2–12 mo; then recovery	1–9 mo; stable at 14 mo	Yes
	Bone marrow	Sideroblastic rings	Erythroblastopenia; vacuolization	NR	Reduced dysplastic erythropoiesis, increased and dysplastic megakaryopoiesis, sideroblastic rings
	Thrombocytopenia	Yes	No	No	NR
Metabolic	Lactic acidosis	No [lactate levels 1.4 mmol/L at 16 y]	Yes (3 mo)	Yes (1 mo)	Yes (13 y)
	Other labs	Low ALP (119 U/L); elevated P (5.9 mg/dL)	NR	NR	High erythropoietin, ferritin and transferrin
Skeletal	Findings	Epiphyseal dysplasia, metaphyseal widening, scoliosis, *genu valgum*, *acetabular–femoral* dysplasia, neoacetabulum formation	None reported	None reported	None reported
Craniofacial	Dysmorphisms	Multiple	NR	NR	NR
Myopathy	Muscle strength/RC enzyme activity	Mild reduced muscle strength and no other symptoms/no RC activity measurement	NAs RC CI/CIII/CIV activity ↓	NAs/NR	Reduced muscle strength and exercise capacity, fatigue, tiredness/NR
Neurology	Clinical exam	[N]	[N]	[N]	[N]
Cardiac	Cardiomyopathy	No	No	No	No
Respiratory	Insufficiency	Absent	NR	NR	NR
Other systems	Ophthalmology	[N]	[N]	NR	NR
Genetic findings	Variant	Homozygous p.(Asp311Glu)	Homozygous p.(Asp311Glu)	Homozygous p.(Asp311Glu)	Homozygous p.(Asp311Glu)
	Domain affected	aACB	aACB	aACB	aACB
Course/ Follow-up	Most recent status	16 y (currently); stable; alive	6 y; good evolution; off transfusions; alive	14 mo; stable, off transfusions; alive	14 y; stable; alive

y: years; NR: not reported; mo: months; aACB: anticodon-binding domain; RC: respiratory chain; [N]: normal; NAs: not clinically assessable.

**Table 2 ijms-26-12039-t002:** p.(Asp311Glu) PV interpretation according to ACMG/AMP.

Criteria	Result	Final Criteria
PS1	The same amino acid change is pathogenic.	Strong
PM2	Extremely low frequency in population databases (gnomAD).	Moderate
PP3	Computational tools unanimously support a deleterious effect on the gene.	Supporting
PS3	A patient harboring the same pathogenic variant exhibited decreased muscle activity of the mitochondrial respiratory chain complexes I, III, and IV, as well as a reduction in maximal respiratory rate in fibroblasts.	Strong
	The ortholog gene:variant [MSY1:p.(Asp333Glu)] in *Saccharomyces cerevisiae* exhibited defective oxidative phosphorylation phenotype.	Supporting

## Data Availability

Data is contained within the article.

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
