# Peer review of "First Latin American Case of MLASA2 Caused by a Pathogenic Variant in the Anticodon-Binding Domain of YARS2"

_ijms, 2025, doi:10.3390/ijms262412039_

Round 1
Reviewer 1 Report
Comments and Suggestions for Authors
Nice and briefly summarized case report of MLASA2 in a Latino patient.
Comments and changes suggested:
- In lines 95/96 of the manuscript, the authors mention about the Italian siblings and it is not immediately apparent that the previous two cases had the same variant and were hence being compared.
- While variable penetrance and expressivity are well established concepts in genetics, it is still not entirely impossible to rule out other unidentified genetic causes of the skeletal findings in this patient (especially given the high ROH% and absence of WGS). The authors should rephrase the abstract, results and discussion to emphasize this point.
- In-silico prediction of protein function is not highly regarded for variant interpretation and the authors must mention how the variant was classified as pathogenic according to ACMG/AMP guidelines.
Author Response
Reviewer 1
Comment 1: In lines 95/96 of the manuscript, the authors mention about the Italian siblings and it is not immediately apparent that the previous two cases had the same variant and were hence being compared.
Response 1: Thank you for your comment. The text was improved in lines 102-105.
Since Ardissone et al. [7] described two Italian siblings harboring the same PV identified in our patient and Riley reported another case [4], we present a comparison in Table 1, allowing a direct comparison of their clinical manifestations [Table 1].
Comment 2: While variable penetrance and expressivity are well established concepts in genetics, it is still not entirely impossible to rule out other unidentified genetic causes of the skeletal findings in this patient (especially given the high ROH% and absence of WGS). The authors should rephrase the abstract, results and discussion to emphasize this point.
Response 2:The reviewer's comment is very accurate. Now, we edited the manuscript as follows:
- Abstract section: The patient presented with sideroblastic anemia and short stature, accompanied by other skeletal dysplasia features not previously associated with MLASA2, including epiphyseal dysplasia, rib edge widening, and poorly defined vertebral structures
- Result section: We acknowledge that we cannot discard all genetic causes of the skeletal findings, but as stated in the clinical presentation section [lines 81-82] “A clinical exome panel focused on skeletal dysplasia was…. negative … ”
- Discussion section lines 180-184: However, we cannot exclude other genetic and environmental factors contributing to the patient's skeletal phenotypes, as no whole-genome sequencing was performed, and we cannot rule out the exposure to other ambient factors that can disrupt growth plate function during the antenatal or postnatal period, because, the limited information associated with the adoption of the patient.
Comment 3: In-silico prediction of protein function is not highly regarded for variant interpretation and the authors must mention how the variant was classified as pathogenic according to ACMG/AMP guidelines.
Response 3: Thank you for the comment, we incorporated a new section in the main text:
2.3 p.(Asp311Glu) PV interpretation according to ACMG/AMP
According to the criteria of the American College of Medical Genetics and Genomics (ACMG) and the Association for Molecular Pathology (AMP), the p.(Asp311Glu) PV fulfills 2 strong criteria (PS1 and PS3), 1 moderate (PM2), and 2 supporting [PP3 and PS3_supporting], classifying it as Pathogenic (Table 2). Functional evidence (PS3) comes from Ardissone's study in which carriers of the pathogenic variant p.(Asp311Glu) showed a reduction in the activity of complexes I, III and IV of oxidative phosphorylation in muscle tissue [7].

Reviewer 2 Report
Comments and Suggestions for Authors
The authors describes the first confirmed MLASA2 case in a Latin American (Mexican) harboring a homozygous pathogenic variant in the YARS2 gene. The patient presented with sideroblastic anemia and short stature, along with skeletal dysplasia features not previously associated with MLASA2, and notably without classic lactic acidosis during childhood. Comprehensive molecular characterization included in silico pathogenicity predictions, protein structural modeling revealing destabilization of the critical KMSKS motif, and runs of homozygosity (ROH) analysis.
I have some commentaries to the authors:
- While in silico analysis is robust, functional validation is absent: 1) No expression studies demonstrating the impact of p.(Asp311Glu) on YARS2 mRNA/protein levels, and 2) No respiratory chain enzyme analysis (though acknowledged by authors). I consider the authors should add a brief discussion of how future functional studies could validate the in silico predictions. Alternatively, cite similar functional validation approaches used for other YARS2 variants.
- While recognizing the clinical complexity, more extensive metabolic profiling would strengthen the case: 1) No plasma or urine organic acids analysis to characterize lactic acidosis more completely (only mentions that lactic acidosis was identified "until adolescence"), 2) Missing mitochondrial marker assessment (acylcarnitine profile, plasma amino acids), and 3) No CoQ10 levels despite mitochondrial dysfunction. I consider the authors should specify exact plasma lactate values and timing of when elevated lactate was detected. If not measured, discuss in context of incomplete metabolic workup.
- While Table 1 comparison with Italian siblings is valuable, deeper mechanistic discussion of why the same p.Asp311Glu variant shows different penetrance/expressivity (timing of anemia onset, presence/absence of lactic acidosis, skeletal involvement) is warranted:
-
Is the skeletal phenotype truly novel for this specific variant, or have other YARS2 variants shown similar features?
-
Why did skeletal features emerge at age 11 but anemia at age 11 yet lactic acidosis delayed to adolescence?
- The manuscript present a limited discussion of natural history variation. The dramatic difference in disease presentation (anemia at 11 years vs. 2 months in Italian cases; no lactic acidosis until adolescence vs. 1-3 months in Italian cases) deserves mechanistic exploration: 1) Could genetic background (Mexican ancestry vs. Italian) modify disease severity?, 2) Role of modifier genes?, and 3) Environmental or nutritional factors?
- If feasible, the authors could obtain muscle biopsy for respiratory chain enzyme analysis and histology (Ragged-Red Fibers, COX negativity). If not feasible, justify why this investigation was deferred and discuss implications for clinical management.
- The Discussion notes that the patient "did not present exercise intolerance, nor was myopathy suspected" initially, yet at age 16 exhibits "decreased muscle strength (4/5 upper, 3/5 lower left extremities)." This suggests subclinical myopathy may be more significant than acknowledged. The authors should clarify whether muscle strength testing was performed at earlier time points. If myopathy is now apparent, discuss whether this represents progression and implications for prognosis.
- While ROH analysis is appropriate and well-executed, the interpretation could be more nuanced. The comparison ROH value in Mexican population (16.05±2.59%) provides mean ± SD, but is the patient's 31.93% statistically significantly different? (Appears yes, but quantification would strengthen). The authors could add statistical testing (e.g., z-score: (31.93-16.05)/2.59 ≈ 6.13, p < 0.001) and cite studies of ROH distribution in definitely consanguineous Mexican families for comparison.
- The DynaMut2 analysis is excellent but presentation could be enhanced. Figure 1 shows the KMSKS motif destabilization but quantitative values for ΔΔG and ΔΔSVib are mentioned in text but not clearly labeled on figure. No comparison to ΔΔG values of other known pathogenic YARS2 variants to establish severity hierarchy. The authors should provide ΔΔG and ΔΔSVib values with units in figure legend or as supplementary data. Cite examples of ΔΔG values for other YARS2 variants if available for comparative context.
- The unusual late presentation of anemia (age 11) and lactic acidosis (adolescence) differs significantly from Italian siblings (anemia at 1-2 months, lactic acidosis at 1-3 months). This temporal pattern deserves more investigation. The authors could discuss possibility of age-dependent penetrance or expressivity and whether similar delayed manifestations have been observed in other YARS2-MLASA2 cases in literature.
- Abstract. Line 22: "other clinical features not previously associated with MLASA" should specify "skeletal dysplasia features" for greater clarityts.
- Introduction. Line 39: OMIM # shows "600462]" with mismatched bracket (should be "]")
- Introduction. Line 39: OMIM number 600462 appears incorrect; standard MLASA2 OMIM is #613561 (referenced elsewhere in text). Verify correct OMIM designation.
- I consider the authors could include a brief overview of other YARS2 variants to contextualize the significance of the anticodon-binding domain location.
- Lines 67-68: "normal lactate dehydrogenase [LDH] (140 U/L)" requires context—what is the lab's reference range? Normal LDH is typically 140-280 U/L, so this is at lower end.
- Line 70-71: "moderate-severe normocytic-normochromic anemia (Hb 7 g/dL)" is significant; current guidelines define moderate anemia as Hb 7-10 g/dL and severe as <7 g/dL, so Hb exactly at 7 g/dL sits at border.
- Line 87: "currently managed with folic acid supplementation alone" is striking but unusual for MLASA2. Has B12 been checked? Any response to other supplementation attempts?
- Add to Methods: "DynaMut2 was run with default parameters including force field energy minimization for 5ns at 300K".
- Consider adding structural comparison: wild-type vs. mutant residue positioning
- Lines 156-158: Recommendation to "consider MLASA2 in infants and children with non-myeloproliferative sideroblastic anemia" is important but generalization is somewhat overstated given single Latin American case.
- Line 168: References p.(Asp333Glu) variant—appears to be typo; should be p.(Asp311Glu)? Verify accuracy
- In Limitations Section, the author could add: "The absence of detailed metabolic profiling (plasma organic acids, carnitine profile) limits complete characterization of mitochondrial dysfunction".
- Line 221: "first Latin-American individual" should be "first Latin American individual".
- Reference 7 (Ardissone et al.) appears twice (refs 7 and 16)—correct citation numbering or consolidate
Comments on the Quality of English LanguageThe manuscript is well-written with clear scientific communication appropriate for an international scientific audience.
There are some minor issues to be corrected.
Author Response
Reviewer 2
Comment 1: While in silico analysis is robust, functional validation is absent: 1) No expression studies demonstrating the impact of p.(Asp311Glu) on YARS2 mRNA/protein levels, and 2) No respiratory chain enzyme analysis (though acknowledged by authors). I consider the authors should add a brief discussion of how future functional studies could validate the in silico predictions. Alternatively, cite similar functional validation approaches used for other YARS2 variants.
Response to Comment 1: We agree. We added in the results and discussion the existence of a functional validation of the p.(Asp311Glu) through the demonstration that a patient harboring this pathogenic variant exhibited decreased muscle activity of the mitochondrial respiratory chain complexes I, III and IV, as well as a reduction in maximal respiratory rate in fibroblasts. Additionally, the ortholog gene:variant [MSY1:p.(Asp333Glu)] in Saccharomyces cerevisiae exhibited defective oxidative phosphorylation phenotype.
Comment 2: While recognizing the clinical complexity, more extensive metabolic profiling would strengthen the case: 1) No plasma or urine organic acids analysis to characterize lactic acidosis more completely (only mentions that lactic acidosis was identified "until adolescence"), 2) Missing mitochondrial marker assessment (acylcarnitine profile, plasma amino acids), and 3) No CoQ10 levels despite mitochondrial dysfunction. I consider the authors should specify exact plasma lactate values and timing of when elevated lactate was detected. If not measured, discuss in context of incomplete metabolic workup.
Response to Comment 2: We acknowledge that the absence of detailed metabolic profiling (plasma organic acids, carnitine profile) limits complete characterization of mitochondrial dysfunction. We added serum lactate concentrations evaluated at age 16 years, which are normal. In addition, we recognize the incomplete metabolic workup as a limitation of the study.
Comment 3 While Table 1 comparison with Italian siblings is valuable, deeper mechanistic discussion of why the same p.Asp311Glu variant shows different penetrance/expressivity (timing of anemia onset, presence/absence of lactic acidosis, skeletal involvement) is warranted.
Response to Comment 3: Of course we discuss deeper the potential mechanisms by which the same p.Asp311Glu variant shows different penetrance/expressivity.
Comment 3.1:Is the skeletal phenotype truly novel for this specific variant, or have other YARS2 variants shown similar features?
Response to Comment 3.1: We clarify that the skeletal dysplasia features (including epiphyseal dysplasia, widening of the metaphysis, irregularity and widening of rib cages, and poorly defined vertebral structures) are novel phenotypes for YARS2 variants and that the only skeletal phenotype previously reported was scoliosis.
Comment 3.2: Why did skeletal features emerge at age 11 but anemia at age 11 yet lactic acidosis delayed to adolescence?
Response to Comment 3.2: Of course, we confirm that skeletal features and anemia were detected at the same time (11 years) with worsening anemia at age 13 requiring two transfusions. Whereas lactic acidosis was discarded and is not present currently at the age of 16 years old.
Comment 3.3: The manuscript presents a limited discussion of natural history variation. The dramatic difference in disease presentation (anemia at 11 years vs. 2 months in Italian cases; no lactic acidosis until adolescence vs. 1-3 months in Italian cases) deserves mechanistic exploration: 1) Could genetic background (Mexican ancestry vs. Italian) modify disease severity?, 2) Role of modifier genes?, and 3) Environmental or nutritional factors?
Response to Comment 3.3: We amplified the discussion regarding the dramatic differences in the clinical presentation between cases with the same pathogenic variant p.Asp311Glu, and discusses mechanistics explorations including the existence of modifier genes, differences in mitochondrial haplogroups, tissue-specific mitochondrial thresholds or nutritional and environmental factors.
Comment 4: If feasible, the authors could obtain muscle biopsy for respiratory chain enzyme analysis and histology (Ragged-Red Fibers, COX negativity). If not feasible, justify why this investigation was deferred and discuss implications for clinical management.
Response to Comment 4: Legal guardians of the patient refused a biopsy and a more in-depth metabolic analysis of the case. However, we recommended to the primary care physician the supplementation with riboflavin and coenzyme Q10, since such drugs improve mitochondrial function and oxidative phosphorylation.
Comment 5: The Discussion notes that the patient "did not present exercise intolerance, nor was myopathy suspected" initially, yet at age 16 exhibits "decreased muscle strength (4/5 upper, 3/5 lower left extremities)." This suggests subclinical myopathy may be more significant than acknowledged. The authors should clarify whether muscle strength testing was performed at earlier time points. If myopathy is now apparent, discuss whether this represents progression and implications for prognosis.
Response to Comment: Appreciated the suggestion. Sure, the patient did not present exercise intolerance since the beginning of its study and currently remains without tiredness or weakness, but we detected decreased muscle strength (4/5 upper, 3/5 lower left extremities) suggesting subclinical myopathy. Consequently, we acknowledge the patient could worse in the next future and that the treatment with riboflavin and coenzyme Q10 might improve muscle function, suggesting its prescription to its primary care physician.
Comment 6: While ROH analysis is appropriate and well-executed, the interpretation could be more nuanced. The comparison ROH value in Mexican population (16.05±2.59%) provides mean ± SD, but is the patient's 31.93% statistically significantly different? (Appears yes, but quantification would strengthen). The authors could add statistical testing (e.g., z-score: (31.93-16.05)/2.59 ≈ 6.13, p < 0.001) and cite studies of ROH distribution in definitely consanguineous Mexican families for comparison.
Response to Comment 6: Sorry, it is not possible to estimate p-value for comparing the ROH value of a single subject (the index case) with the cohort nor estimate SD, but we mention in the discussion that two indigenous groups in Mexico, who present high consanguinity and inbreeding, have the highest ROH values. You could find it in the supplementary figure 9 of the reference 13. But, we didn’t add specific values for ethical concerns since we do not want them to be labeled as consanguineous and endogamous. This was even the recommendation of the editors and reviewers of the article cited as 13.
Comment 7: The DynaMut2 analysis is excellent but the presentation could be enhanced. Figure 1 shows the KMSKS motif destabilization but quantitative values for ΔΔG and ΔΔSVib are mentioned in text but not clearly labeled on figure. No comparison to ΔΔG values of other known pathogenic YARS2 variants to establish severity hierarchy. The authors should provide ΔΔG and ΔΔSVib values with units in figure legend or as supplementary data. Cite examples of ΔΔG values for other YARS2 variants if available for comparative context.
Response to Comment 7: Of course, in lines 131-134 we added the ΔΔG values for other pathogenic variants reported by Sommerville 2017 and Riley 2018. “The ΔΔG for other pathogenic variants such p.(Gly46Asp) [catalytic domain], p.(Cys369Tyr) [anticodon-binding domain] and Gly244Ala [catalytic domain] are -0.84 kcal/mol, -0.82kcal/mol, and -0.52kcal/mol, respectively.”
Comment 8: The unusual late presentation of anemia (age 11) and lactic acidosis (adolescence) differs significantly from Italian siblings (anemia at 1-2 months, lactic acidosis at 1-3 months). This temporal pattern deserves more investigation. The authors could discuss the possibility of age-dependent penetrance or expressivity and whether similar delayed manifestations have been observed in other YARS2-MLASA2 cases in literature.
Response to Comment 8: Thank you, we discuss the age -dependent expressivity and mechanistics explorations including the existence of modifier genes, differences in mitochondrial haplogroups, tissue-specific mitochondrial thresholds or nutritional and environmental factors. We also mention that we cannot discard the future presentation of clinical recognizable myopathy and lactic acidosis.
Comment 9: Abstract. Line 22: "other clinical features not previously associated with MLASA" should specify "skeletal dysplasia features" for greater clarityts.
Response to Comment 9: Thank you, we did it.
Comment 10:Introduction. Line 39: OMIM # shows "600462]" with mismatched bracket (should be "]")
Response to Comment 10: Corrected
Comment 11: Introduction. Line 39: OMIM number 600462 appears incorrect; standard MLASA2 OMIM is #613561 (referenced elsewhere in text). Verify correct OMIM designation.
Response to Comment 11: You are correct. We edited the text.
Comment 12: I consider the authors could include a brief overview of other YARS2 variants to contextualize the significance of the anticodon-binding domain location.
Response to Comment 12: In a systematic review and metanalysis currently under consideration for publication in another journal of MDPI, we present and analyse all pathogenic variants in YARS2.
Comment 13: Lines 67-68: "normal lactate dehydrogenase [LDH] (140 U/L)" requires context—what is the lab's reference range? Normal LDH is typically 140-280 U/L, so this is at lower end.
Response to Comment 13: Thank you, we clarified it.
Comment 14: Line 70-71: "moderate-severe normocytic-normochromic anemia (Hb 7 g/dL)" is significant; current guidelines define moderate anemia as Hb 7-10 g/dL and severe as <7 g/dL, so Hb exactly at 7 g/dL sits at border.
Response to Comment 14: Sure, at the lower border. We specified it.
Comment 15: Line 87: "currently managed with folic acid supplementation alone" is striking but unusual for MLASA2. Has B12 been checked? Any response to other supplementation attempts?
Response to Comment 15: We confirm that no other supplements have been administered to date. We have added the possible role of folic acid to the discussion, even though a couple of patients with distinct pathogenic variants in YARS2 have not responded to folate treatment.
Comment 16: Add to Methods: "DynaMut2 was run with default parameters including force field energy minimization for 5ns at 300K".
Response to Comment 16: Appreciated.
Comment 17: Consider adding structural comparison: wild-type vs. mutant residue positioning.
Response to Comment 17: Perfect! We improved and presented the structural comparison between wild-type Asp311 and mutant Glu311. Figure 1. Structural comparison of the wild-type Asp311 and mutant Glu311 residues and their impact on the KMSKS motif of YARS2. Left: wild-type Asp311 (D311, green) within the anticodon-binding domain, showing its spatial relationship with the KMSKS loop (cyan) and neighboring residues His280, Arg287, and Gln290. Right: mutant Glu311 (311E, green) modeled at the same coordinates, displaying changes in side chain that alters local packing and the interaction network around the KMSKS motif. DynaMut2 predicts a destabilizing effect for the p.Asp311Glu substitution with ΔΔG = −0.600 kcal/mol and an increase in molecular flexibility with ΔΔSVib ENCoM = +0.663 kcal·mol⁻¹·K⁻¹.
Comment 18: Lines 156-158: Recommendation to "consider MLASA2 in infants and children with non-myeloproliferative sideroblastic anemia" is important but generalization is somewhat overstated given a single Latin American case.
Response to Comment 18: Agree. We eliminated such affirmation
Comment 19: Line 168: References p.(Asp333Glu) variant—appears to be a typo; should be p.(Asp311Glu)? Verify accuracy
Response to Comment 19: It is accurate because such variant is the ortholog variant/gene in Saccharomyces cerevisiae [MSY1:p.(Asp333Glu)] according to the Reference 7.
Comment 20: In the Limitations Section, the author could add: "The absence of detailed metabolic profiling (plasma organic acids, carnitine profile) limits complete characterization of mitochondrial dysfunction".
Response to Comment 20: Added
Comment 21: Line 221: "first Latin-American individual" should be "first Latin American individual".
Response to Comment 21: Corrected
Comment 22: Reference 7 (Ardissone et al.) appears twice (refs 7 and 16)—correct citation numbering or consolidate.
Response to Comment 22: Corrected
